# Identification of Cryptic Species of Four *Candida* Complexes in a Culture Collection

**DOI:** 10.3390/jof5040117

**Published:** 2019-12-17

**Authors:** Gustavo Fontecha, Kathy Montes, Bryan Ortiz, Celeste Galindo, Sharleen Braham

**Affiliations:** 1Microbiology Research Institute, Universidad Nacional Autónoma de Honduras, Tegucigalpa 11101, Honduras; kathy.montes@unah.hn (K.M.); bryanortiz_02@hotmail.com (B.O.); celestegalindom@gmail.com (C.G.); sharbraham@icloud.com (S.B.); 2Instituto Hondureño de Seguridad Social, Tegucigalpa 11101, Honduras

**Keywords:** *Candida* spp., cryptic species, Honduras, PCR-RFLP, *hwp1* gene, *gpi* gene, *C. auris*

## Abstract

*Candida* spp. are one of the most common causes of fungal infections worldwide. The taxonomy of *Candida* is controversial and has undergone recent changes due to novel genetically related species. Therefore, some complexes of cryptic species have been proposed. In clinical settings, the correct identification of *Candida* species is relevant since some species are associated with high resistance to antifungal drugs and increased virulence. This study aimed to identify the species of four *Candida* complexes (*C. albicans*, *C. glabrata*, *C. parapsilosis*, and *C. haemulonii*) by molecular methods. This is the first report of six cryptic *Candida* species in Honduras: *C. dubliniensis, C. africana, C. duobushaemulonii, C. orthopsilosis,* and *C. metapsilosis,* and it is also the first report of the allele *hwp1-2* of *C. albicans sensu stricto*. It was not possible to demonstrate the existence of *C. auris* among the isolates of the *C. haemulonii* complex. We also propose a simple method based on PCR-RFLP for the discrimination of the multi-resistant pathogen *C. auris* within the *C. haemulonii* complex.

## 1. Introduction

*Candida* species are one of the most important pathogens in medical mycology [1], and the incidence of candidiasis in recent years seems to be on the rise [2,3]. These microorganisms are a frequent cause of bloodstream [4] and vulvovaginal infections [5]. Some species have also been described as important nosocomial pathogens in newborns [6], the elderly [7], transplant recipients [8], and immunocompromised patients [9], among others. *Candida albicans* is the most virulent and commonly isolated species from clinical samples, even though the reports of non-*albicans* species are increasing [10,11,12]. Candidiasis is mainly caused by four *Candida* species comprising *C. albicans*, *C. glabrata*, *C. parapsilosis*, and *C. tropicalis* [13,14].

The taxonomy of some species of the genus *Candida* has undergone recent changes due to the description of new genetically related species. Given the difficulties in determining their precise identification, they are currently grouped in complexes of cryptic species [15]. The *C. albicans* complex includes *C. albicans sensu stricto (s.s.)*, *C. dubliniensis* [16], and *C. africana* [17]. The *Candida glabrata* complex comprises *C. glabrata s.s.*, *C. bracarensis* [18], and *C. nivariensis* [19]. *Candida parapsilosis s.s., C. orthopsilosis,* and *C. metapsilosis* form the complex *C. parapsilosis* [20]. A fourth complex of cryptic species includes *C. haemulonii s.s.*, *C. haemulonii* var. *vulnera*, and *C. duobushaemulonii* [21]. Other phylogenetically closely related species to the *C. haemulonii* complex have been described, namely *C. pseudohaemulonii, C. ruelliae, C. heveicola*, and *C. auris* [22,23].

The correct identification of cryptic species in a clinical setting is relevant from an epidemiological and medical point of view. It is also important to better understand the evolution of antifungal resistance. Perhaps the most notable example of the importance of identifying cryptic species is the appearance and rapid dispersion of *C. auris*. This species is considered a serious threat to public health worldwide due to frequent relapses and treatment failures [24,25]. The emergence of new cryptic species of *Candida* poses a challenge for clinical laboratories because it is not always possible to have updated methodologies for their correct identification, particularly in low-income countries. Conventional methods based on the assimilation of carbohydrates or chromogenic media are designed to identify the most common species of yeasts, but do not detect all cryptic species. The introduction of matrix-assisted laser desorption ionisation time-of-flight mass spectrometry (MALDI-TOF-MS) in the clinical laboratory has considerably improved the identification of fungi [26]. As a result of the introduction of this approach, it is possible to identify most cryptic species, however, this technology is still expensive, requires constant database updates, and its use is restricted to high-income countries or third-/fourth-level hospitals.

A growing list of molecular methods for the specific discrimination of cryptic *Candida* species has been proposed [15,27,28,29,30,31]. These molecular methods are intended to be cost-effective alternatives for implementation into routine microbiological diagnosis. However, the disadvantage is that they have not been commercially standardized or approved by regulatory agencies. With increased demand, the implementation of a fast and simple method based on real-time multiplex PCR may be available to most clinical laboratories at a reasonable cost in the near future [32]. Meanwhile, and due to the lack of information in Honduras about cryptic species circulating in the country, this study aimed to identify the species of four *Candida* complexes by five molecular methods. We also describe a simple method based on PCR-RFLP for the discrimination of *C. auris* within the *C. haemulonii* complex.

## 2. Materials and Methods

### 2.1. Yeast Isolates and Phenotypic Identification

This study analyzed 108 yeast isolates from a collection of fungi previously obtained from different clinical samples such as urine, sputum, vaginal swabs, blood, catheters, stool/rectal swabs, cutaneous secretion, otic secretion, oral swabs, and cerebrospinal fluid (CSF) [33]. Only cryptic *Candida* species that belonged to one of the four common complexes were included: *C. albicans* (*n* = 66), *C. glabrata* (*n* = 24), *C. parapsilosis* (*n* = 15)*,* and *C. haemulonii* (*n* = 3). Yeasts species were identified by a phenotypic approach and five simple molecular methods. The first identification method was based on a phenotypic approach through the chromogenic medium HardyCHROM^®^ (CRITERION^®^, Hardy Diagnostics, Santa Maria, CA, USA). According to the manufacturer, colonies of the *C. albicans* complex showed a dark metallic green color; *C. glabrata* complex produced medium size, smooth, pink colored colonies, often with a darker mauve center; while the dry and dark purple colonies were assigned to *C. parapsilosis* complex. Other species produced small, white-to-pink colored colonies. The species of the *C. haemulonii* complex were only identified by molecular approaches since the chromogenic method is not a robust approach for identifying species within this complex [34].

### 2.2. PCR-RFLP of the ITS Region and MspI

DNA from the axenic cultures was extracted following a previously described protocol using a BeadBeater^®^ system (Bio Spec products Inc., Bartlesville, OK, USA) with glass beads, and phenol-chloroform [35]. The four *Candida* complexes were identified by PCR-RFLP, amplifying the internal transcribed sequence (ITS) of the ribosomal region and performing DNA digestion using the enzyme MspI [35,36]. According to this method, the species within the complexes *C. albicans*, *C. parapsilosis*, and *C. haemulonii* cannot be differentiated from each other, since they produce similar restriction patterns. All species of the *C. albicans* complex produced a restriction pattern of 299 and 239 bp; the four species of the *C. haemulonii* complex produced a pattern of 374–403 bp; and the three species of the *C. parapsilosis* complex produced a unique 530 bp band; whereas, the three species of the *C. glabrata* complex produce three characteristic restriction profiles (*C. glabrata*: 563/318 bp, *C*. *nivariensis*: 319/238/207 bp, *C. bracarensis*: 550/255 bp). Therefore, this is the only *Candida* out of the four complexes whose species can be identified by this method.

PCR conditions were carried out in a volume of 50 µL, with 25 µL of 2× PCR Master Mix (Promega Corp. Madison, WI, USA), 1 µL of each primer at 10 µM, 22 µL of water, and 1 µL of DNA (40 ng/µL). The primers used were: ITS1: 5′-TCCGTAGGTGAACCTGCGG-3′ and ITS4: 5′-TCCTCCGCTTATTGATATGC-3′ The amplification program was as follows: 95 °C for 5 min, 37 cycles of 95 °C for 30 s, 56 °C for 30 s, and 72 °C for 30 s, and an extension at 72 °C for 5 min. After confirming the amplification, 10 µL of the amplification product was digested with 0.5 µL of MspI (10 U/µL) at 37 °C for 2 h, 2 µL of buffer, and 0.2 µL of 10 µg/µL acetylated bovine serum albumin (BSA), for 2 h at 37 °C. The PCR products and digested PCR products were separated and visualized on a 1.5% agarose gel with ethidium bromide. A working algorithm of the molecular methods used in this study can be seen in Figure 1.

### 2.3. PCR of the hwp1 Gene for Identification of Species from C. albicans Complex

A method used to differentiate cryptic species within the *C. albicans* complex was based on the amplification of the hyphal wall protein 1 gene (*hwp1*) [37]. This method was originally proposed to discriminate *C. africana* from the other species of the *C. albicans* complex through length polymorphisms within these sequences. The three species generate different band sizes: *C. africana* (750 bp), *C. albicans* (941 and 850 bp), and *C. dubliniensis* (569 bp) [5]. DNA was amplified in a 50 µL volume, with 25 µL of 2X PCR Master Mix, 2 µL of each primer at 10 µM, 19 µL of water, and 2 µL of DNA (40 ng/µL). The sequences of the primers were: CR-f: 5′-GCTACCACTTCAGAATCATCA TC-3′ and CR-r: 5′-GCACCTTCAGTCGTAGAGACG-3′. PCR reaction conditions were as follows: denaturation at 95 °C for 5 min, 35 cycles of denaturation at 94 °C for 45 s, annealing at 58 °C for 40 s, and extension at 72 °C for 55 s, with a final extension at 72 °C for 10 min. All the PCR products and digested PCR products were separated and visualized on a 1.5% agarose gel with ethidium bromide.

### 2.4. PCR-RFLP for the Identification of C. auris within the C. haemulonii Complex

Because the species of the *C. haemulonii* complex cannot be identified by amplification of the ITS region and digestion with MspI, a simple method was designed to differentiate *C. auris* from other species phylogenetically related to the *C. haemulonii* complex: *C. haemulonii*, *C. haemulonii* var. *vulnera, C. duobushaemulonii, C. pseudohaemulonii, C. ruelliae,* and *C. heveicola* [23]. The ITS region was amplified for subsequent digestion with the restriction enzyme AluI. To assess whether the restriction target for AluI was a conserved site, sequences available in NCBI of each of the seven species were analyzed. DNA sequences of *C. haemulonii* complex species isolated from countries on four continents were obtained (Americas, Europe, Asia, and Oceania). According to the in silico analysis (Figure 2), *C. auris* exhibited one or two restriction targets for the AluI enzyme and generates a two- or three-band restriction profile (305/95 bp; or 264/98/36 bp), while the rest of the species showed a single-band pattern between 374 and 392 bp. The amplification conditions were the same as described in the Section 2.2. Ten microliters of the amplification products of the ITS region were digested in a volume of 20 µL, with 0.5 µL of the AluI enzyme (10 U/µL), 2 µL of buffer, 17.3 µL of water, and 0.2 µL of acetylated bovine serum albumin (BSA) (10 µg/µL) for 2 h at 37 °C. A previously characterized strain of *C. auris* was used as a control in all assays.

### 2.5. PCR of the gpi Gene for Identification of C. auris

A PCR assay described for the rapid identification of *C. auris* was also performed on the isolates of the *C. haemulonii* complex. This technique amplified a unique glycosylphosphatidylinositol (GPI) protein-encoding gene originally described as QG37_03410 [38]. A standard PCR mix was prepared with 12.5 µL of 2× PCR Master Mix, 1 µL of each primer: 03410_F: 5′-GCCGCTAGATTGATCACCGT-3′ and 03410_R: 5′-TAGGTGTGGGTACCCTTGGT-3′, 9 µL of water, and 1.5 µL of DNA template. The PCR program consisted of 1 cycle at 94 °C for 3 min, 35 cycles of 30 s at 94 °C, 30 s at 60 °C, 30 s at 72 °C, and a final cycle at 72 °C for 3 min. A size band of 137 bp was expected for *C. auris* and no amplification for the rest of the species of the *C. haemulonii* complex. A reference strain was used as control.

### 2.6. Sequencing of PCR Products

To confirm the correct identification of cryptic species of the four complexes, five amplification products of the ITS region and four sequences of the *hwp1* gene were sequenced on both strands using the same pair of primers used for amplification. Sequencing services were provided by Psomagen^®^ (www.macrogenusa.com). The sequences were edited with the Geneious^®^ 9.1.7 software and deposited into the NCBI GenBank.

## 3. Results

A total of 108 isolates were examined in this study. All isolates were previously deposited in the yeast collection of the National University of Honduras and were identified as cryptic species of one of the four complexes: *C. albicans* (*n* = 66), *C. glabrata* (*n* = 24), *C. parapsilosis* (*n* = 15), and *C. haemulonii* (*n* = 3) (Table 1). Phenotypic identification was based on the characteristics of the colony in chromogenic medium and the isolates were subsequently identified by PCR-RFLP by digesting the amplified fragment of the ITS ribosomal region with MspI (Figure 3a). The isolates were obtained from ten different types of clinical samples—mainly urine, sputum, and vaginal swabs.

### 3.1. Identification of Cryptic Species within the C. albicans Complex

*Candida* species that were identified as *C. albicans* complex by PCR-RFLP were also identified by a length polymorphism in the *hwp1* gene. Of the 66 isolates, 52 (78.8%) were *C. albicans sensu stricto* with a 941 bp homozygous allele *hwp1*-*1*, 3 (4.5%) isolates were *C. dubliniensis*, 2 (3%) were *C. africana*, and one isolate showed an homozygous form of *C. albicans* with the rare allele *hwp1*-*2* (1.5%), described in 2009 [39]. Eight isolates (12.1%) showed a two-band pattern compatible with a heterozygous strain of *C. albicans* (Figure 3d). In total, 61 (92.4%) of the isolates were identified as *C. albicans sensu stricto* regardless of the *hwp1* gene allele, and 5 (7.6%) were identified as non-*albicans* cryptic species. To our knowledge, this is the first study that reports *C. dubliniensis*, *C. africana*, and a heterozygous *hwp1* variant of *C. albicans* in Honduras. *Candida dubliniensis* isolates were obtained from sputum or pharyngeal exudates. Both isolates of *C. africana* came from vaginal swabs and urine, while the eight strains with a heterozygous profile were isolated from urine, sputum, oral swab, and vaginal swab.

Two specimens of each species were randomly selected and the fragments of the *hwp1* gene were sequenced. Individuals with a two-band pattern (*C. albicans*, *hwp1-1/hwp1*-*2*) could not be sequenced. The identity of all isolates was confirmed through the NCBI BLAST tool. The sequences obtained were deposited in Genbank with the accession numbers: MN719370-3.

### 3.2. Identification of Cryptic Species of the C. haemulonii Complex

Three isolates showed an amplicon of approximately 400 bp when amplifying the ITS region and none of them showed restriction fragments with target sites for the MspI enzyme. This pattern was consistent with the *C. haemulonii* complex. To identify these species, three molecular methods were used. The first method was based on the digestion with AluI of the ITS region. According to an extensive in silico analysis, only *C. auris* has restriction target site for AluI, unlike the other six species (*C. haemulonii*, *C. haemulonii* var. *vulnera*, *C. duobushaemulonii*, *C. pseudohaemulonii*, *C. ruelliae*, and *C. heveicola*) that do not have a restriction target site for the enzyme to cleave and therefore maintain a band of around 400 bp. According to this analysis, none of the three isolates were identified as *C. auris*. A reference strain was used as a positive control in all assays (Figure 3b). The second method used to identify *C. auris* was a specific amplification of the *gpi* gene [38]. This method did not reveal the presence of *C. auris* among the three isolates (Figure 3c). The ITS regions of the three isolates were sequenced and all were identified as *C. duobushaemulonii* (accession number: MN699480). These species were isolated from two vaginal swabs and a cutaneous secretion. This is the first report of *C. duobushaemulonii* being identified in Honduras.

### 3.3. Identification of Cryptic Species of the C. glabrata and C. parapsilosis Complexes

All of the 24 isolates identified as *C. glabrata* complex showed a restriction pattern compatible with *C. glabrata sensu stricto* (Figure 3a). Sequencing of the ITS region confirmed this result (accession number MN699479). Therefore, it was not possible to identify *C. nivariensis* or *C. bracarensis* in this study. Most of the *C. glabrata* isolates were derived from urine samples (66.7%) (Table 1). With respect to the 15 isolates of the *C. parapsilosis* complex, the sequencing results of the ITS region showed that 11 were *C. parapsilosis s.s.*, three were *C. orthopsilosis* and one isolate was identified as *C. metapsilosis* (accession numbers MN699481-3). This would also be the first report of the two less-frequent cryptic species of the *C. parapsilosis* complex in Honduras.

## 4. Discussions

Identification of yeasts at the species level is a challenge for clinical microbiologists who work for health centers in low-income countries due to the lack of appropriate technologies. Conventional methods based on phenotypic or biochemical characteristics are sometimes insufficient to provide an accurate identification of the etiological agent of an infection [40,41]. Reports of novel *Candida* species could be clinically relevant because they could differ both in virulence and in the spectrum of antifungal resistance [42]. Consequently, the lack of specific microbiological data could force physicians to empirically treat life-threatening mycoses with broad-spectrum antifungal medications, which would impact existing issues with antifungal resistance. Therefore, this study aimed to identify archived cryptic specimens belonging to four complexes of the genus *Candida* isolated from clinical samples from a hospital in Honduras in order to demonstrate the existence of novel species that the laboratory does not routinely identify and report. One hundred and eight isolates of the *C. albicans*, *C. glabrata*, *C. parapsilosis*, and *C. haemulonii* complexes were analyzed by molecular methods.

*C. albicans* complex spp. are the most frequently isolated from clinical samples [35,43]. However, cryptic species within this complex, such as *C. dubliniensis* and *C. africana*, are routinely misidentified or unidentified. Thus, this bias could slightly overestimate the number of *C. albicans* that is reported [5,44,45,46]. In this study, 66 isolates of the albicans complex were analyzed by amplification of the *hwp1* gene and the result was confirmed by sequencing. More than 92% of the individuals were identified as *C. albicans sensu stricto,* and 7.6% were identified as *C. dubliniensis* or *C. africana*. This proportion is higher than reported in similar studies. A study from Argentina analyzed isolates obtained from patients with vulvovaginal candidiasis and showed that only two infections (3.8%) were caused by *C. dubliniensis*. *Candida africana* was not found among those isolates [5]. In Turkey, the reexamination of 376 yeasts obtained from vaginal infections showed 1.6% prevalence of *C. dubliniensis* and *C. africana* [44]. Another study carried out in China with more than one thousand isolates showed 1.5% prevalence of *C. africana* but did not report *C. dubliniensis* [47].

An interesting result of our study is the finding of 12.12% of heterozygous strains (*hwp1*-*1/hwp1*-*2*) and 1.5% of homozygous strains (*hwp1*-*2/hwp1*-*2*) of *C. albicans*. A shorter allele of 850 bp (*hwp1*-*2*) was initially described in 2009 [39] as a novel version of the wild-type allele of 941 bp (*hwp1*-*1*), and three deletions were described within the gene. Since then, there have been numerous reports of this genotype worldwide [5,47,48]. More recently, authors in Cameroon reported five genotypes composed of the combination of four alleles of the *hwp1* gene (941, 1080, 850, and 1200 bp) [49]. Hence, our results support the high diversity of this gene in *C. albicans*, but despite these polymorphisms, the *hwp1* gene has been shown to be a good marker for the identification of cryptic species of this complex. As far as we know, this would be the first report of *C. dubliniensis* and *C. africana* in Honduras. Both strains of *C. africana* were isolated from genital samples. This is concordant with that described by other authors [48,50,51]. Although some authors argue that there are no compelling reasons to separate *C. dubliniensis* or *C. africana* from *C. albicans* routinely [52,53], some studies show that the susceptibility of non-*albicans* species to antifungals could be lower than that of *C. albicans* [54]. In addition to the implications for clinical practice, the correct identification of cryptic species contributes to the knowledge of the epidemiology of candidiasis in Latin America [55].

One of the main reasons that motivated this study was the search for *C. auris* in the country, where it has not yet been reported. *Candida auris* is currently considered as one of the most important emerging microorganisms for public health, due to its virulence, its multiple resistance against different classes of antifungal drugs and high mortality rate [25]. There are reports of this yeast in neighboring countries such as Panama [56,57] and Colombia [58], and consequently there are high probabilities that this species will be dispersed to the rest of Central America in the near future. Phenotypic methods available in Honduras would not allow the identification of *C. auris* if it were to occur. Especially as its pattern of amplification of the ribosomal region is 400 bp, which resembles that of other species of the *C. haemulonii* complex [35]. In this study, we analyzed three yeasts with a restriction pattern congruent with *C. haemulonii* complex. Species identification was carried out by specific amplification of the *gpi* gene, and a simple PCR-RFLP technique was designed that allows *C. auris* to be discriminated from the rest of the species of the *haemulonii* complex. Both approaches were concordant that none of the three strains were *C. auris*. Sequencing of the ribosomal region revealed that the three isolates were *C. duobushaemulonii*. In the short term, it will be necessary to expand the search for *C. auris* with a greater number of strains.

It is rare to isolate species of the *C. haemulonii* complex from clinical samples. A study in Brazil identified 40 (0.3%) strains of the *C. haemulonii* complex from a collection of 14,632 isolates [59]. However, a considerable number of isolates suspected to be *C. auris* in Panama were further identified as *C. duobushaemulonii* and some of them were responsible for bloodstream infections and showed elevated minimal inhibitory concentrations for fluconazole, voriconazole, and amphotericin B [56]. Other Brazilian studies reported *C. haemulonii* complex isolates with low susceptibility to fluconazole or amphotericin B [60,61]. These data suggest that although these species are rare, they could be clinically relevant, and their correct identification is recommended.

The third complex evaluated was *C. glabrata*. Twenty-four isolates of *C. glabrata sensu lato* were analyzed by PCR-RFLP and sequencing of the ribosomal ITS and all the strains were identified as *C. glabrata sensu stricto*. Several studies have investigated the presence of rare cryptic species of the *glabrata* complex and have not been able to prove their presence [62,63,64]. However, other studies have reported *C. nivariensis* and *C. bracarensis* but with very low prevalences [28,65,66]. In relation to the 15 strains of the *C. parapsilosis* complex, the ITS sequence showed that the majority of the isolates were *C. parapsilosis s.s.*, wherein three were identified as *C. orthopsilosis* and one of them was *C. metapsilosis*. It seems that the distribution of these cryptic species is also consistent with what was reported before in some countries [62,67,68,69]. This is the first description of cryptic species of the *C. parapsilosis* complex in Honduras, and since these species are frequently associated with invasive infections and exhibit differences in susceptibility to antifungals [70], this information will be useful to improve the species-specific identification of *Candida*.

## 5. Conclusions

This study has demonstrated the presence of cryptic species from four *Candida* complexes that are not routinely reported in Honduras’s hospitals. Additionally, the presence of *C. dubliniensis*, *C. africana*, *C. duobushaemulonii*, *C. orthopsilosis*, *C. metapsilosis*, and the *hwp1*-*2* allele of *C. albicans* is described for the first time in Honduras. At present, due to the limitation of molecular technologies in Honduras, a simple method based on PCR-RFLP with AluI restriction enzyme is proposed for the identification of *C. auris* among other species of the *C. haemulonii* complex.

## Figures and Tables

**Figure 1 jof-05-00117-f001:**
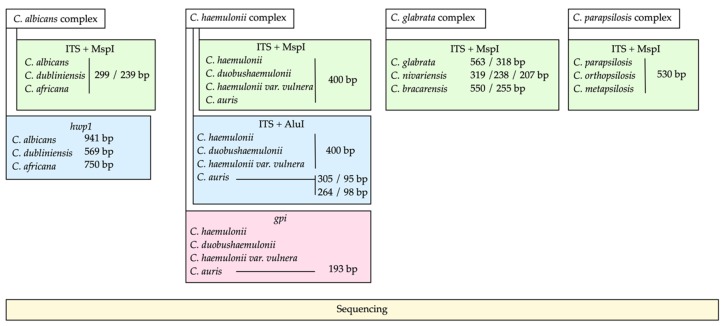
Flow diagram for the identification of cryptic species within four complexes of the genus *Candida* by molecular methods.

**Figure 2 jof-05-00117-f002:**
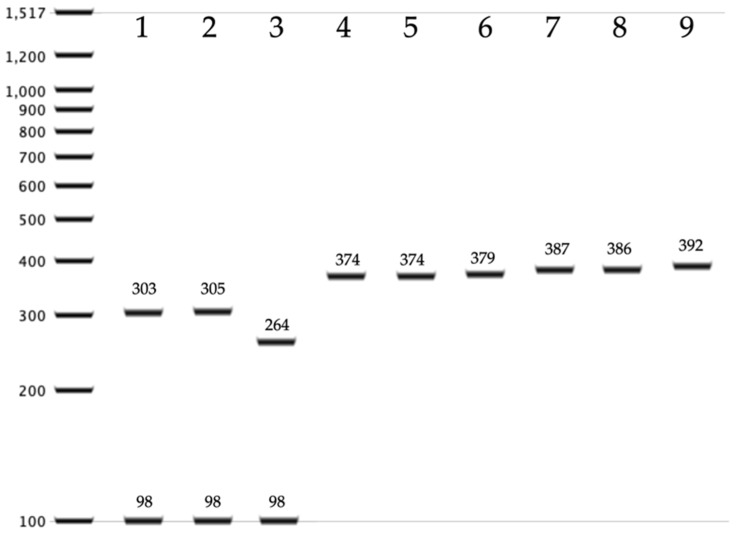
In silico analysis of a PCR-RFLP method of the ITS region digested with AluI for identification of species from *Candida haemulonii* complex. Lanes 1–3: *Candida auris*; Lanes 4,5: *C. haemulonii* and *C. haemulonii* var. *vulnera*; Lane 6: *Candida ruelliae*; Lane 7: *Candida duobushaemulonii*; Lane 8: *Candida pseudohaemulonii*; Lane 9: *Candida heveicola*.

**Figure 3 jof-05-00117-f003:**
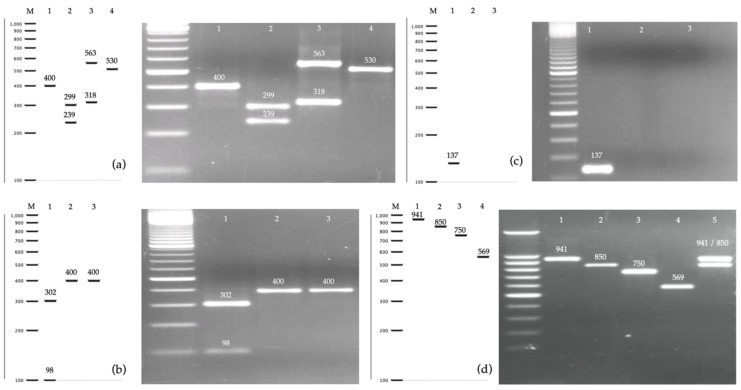
In silico analysis and amplification profiles of four molecular methods for the identification of *Candida* species. (**a**) PCR of the ITS region of rDNA digested with MspI. Lane 1: *C. haemulonii* complex, Lane 2: *C. albicans* complex, Lane 3: *C. glabrata sensu stricto*, Lane 4: *C. parapsilosis* complex; (**b**) PCR of the ITS region of rDNA digested with AluI for identification of *C. auris* within the *C. haemulonii* complex. Lane 1: *C. auris*, Lanes 2,3: *C. haemulonii*; (**c**) PCR of the *gpi* gene for identification of *C. auris* within the *C. haemulonii* complex. Lane 1: *C. auris*, lanes 2,3: *C. haemulonii;* (**d**) PCR of the *hwp1* gene for identification of species within the *C. albicans* complex. Lane 1: *C. albicans sensu stricto hwp1-1*, Lane 2: *C. albicans hwp1-2*, Lane 3: *C. africana*, Lane 4: *C. dubliniensis*, Lane 5: heterozygous isolate showing a two-band profile *hwp1-1/hwp1-2*.

**Table 1 jof-05-00117-t001:** Cryptic species of *Candida* according to clinical sample and identified by PCR-RFLP of the ITS region and MspI.

Clinical Sample	No. of Samples (%)	*C. albicans* Complex	*C. glabrata* Complex	*C. parapsilosis* Complex	*C. haemulonii* Complex
Urine	38 (35.19)	21	16	1	-
Sputum	28 (25.93)	25	1	2	-
Vaginal swab	16 (14.81)	10	4	-	2
Blood	7 (6.48)	2	-	5	-
Catheter	7 (6.48)	3	1	3	-
Stool	2 (1.85)	-	-	2	-
Cutaneous secretion	5 (4.63)	1	2	1	1
Otic secretion	1 (0.93)	-	-	1	-
Oral swab	2 (1.85)	2	-	-	-
Cerebrospinal fluid (CSF)	2 (1.85)	2	-	-	-
Total (%)	108 (100%)	66 (61.11%)	24 (22.22%)	15 (13.88%)	3 (2.78%)

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
