# Peer review of "Identification of Cryptic Species of Four Candida Complexes in a Culture Collection"

_jof, 2019, doi:10.3390/jof5040117_

Round 1

Reviewer 1 Report

The authors of jof-663334 manuscript aimed to identify cryptic species within four Candida complexes. The manuscript is well written and provides new insights into the Candida species occurrence in Honduras.

MAJOR COMMENTS
Materials and Methods
It is not clear why only a few ITS fragments were sequenced. This should be performed for all the isolates and serve as an ID reference.

MINOR COMMENTS
the whole manuscript - genus and species names should be written in italics

Line 26: should be "one of the most important pathogens" not "the most important pathogens"

Line 27: the authors say "in recent years..." and then cite a paper from 2006 [2] - a newer reference should be added

Line 32: newer references should be added to support the claim of changing species distribution - see Pfaller et al 2019 "Twenty Years of the SENTRY Antifungal Surveillance Program: Results for Candida Species From 1997-2016"

Author Response

Point by point response
Reviewer 1

MAJOR COMMENTS

Materials and Methods. It is not clear why only a few ITS fragments were sequenced. This should be performed for all the isolates and serve as an ID reference.

Answer: We appreciate the comment made by the reviewer. Indeed, sequencing of all strains would have been desirable; however, the purpose of sequencing the fragments was only confirmatory to validate the identification and profiles obtained with the other molecular methods (PCR-RFLP, PCR). In addition, in the previous experiences in our laboratory the species found have exactly the same sequences and without polymorphisms among them. Therefore, we considered it unnecessary to sequence all strains.

MINOR COMMENTS

the whole manuscript - genus and species names should be written in italics.

Answer: All scientific names and genes have been reviewed and written in italics.

Line 26: should be "one of the most important pathogens" not "the most important pathogens"

Answer: The sentence has been modified according to the reviewer's recommendation: “Candida species are one of the most important pathogens in medical mycology”

Line 27: the authors say "in recent years..." and then cite a paper from 2006 [2] - a newer reference should be added

Answer: Two new citations have been added to update this idea (Chapman, B. et al. Changing epidemiology of candidaemia in Australia. J Antimicrob Chemother 2017, 72, 1270, doi:10.1093/jac/dkx047., and Quindos, G. Epidemiology of candidaemia and invasive candidiasis. A changing face. Rev Iberoam Micol 2014, 31, 42-48, doi:10.1016/j.riam. 2013.10.001.) The old reference of 2006 has been removed.

Line 32: newer references should be added to support the claim of changing species distribution - see Pfaller et al 2019 "Twenty Years of the SENTRY Antifungal Surveillance Program: Results for Candida Species From 1997-2016"

Answer: A new and more recent reference (Pfaller et al 2019) has been included here.

Reviewer 2 Report

The manuscript by Fontecha and colleagues is a retrospective study aimed at identifying cryptic species of four Candida complexes, namely C. albicans, C. glabrata, C. parapsilosis, and C. haemulonii, in a collection of strains previously collected from a tertiary-level hospital in Tegucigalpa, Honduras. The authors use different molecular methods, and report the presence of cryptic species that were not previously identified in Honduras (i.e . C. dubliniensis, C. africana, C. duobushaemulonii, C. orthopsilosis, and C. metapsilosis).

Highlight:

The authors improve the body of knowledge about cryptic Candida species circulating in Honduras, and they do so by using molecular methods that are widely accessible and can be implemented into microbiological diagnosis laboratories in a cost effective way. Notably, they report for the first time cryptic species that were not previously identified in Honduras (i.e. C. dubliniensis, C. africana, C. duobushaemulonii, C. orthopsilosis, and C. metapsilosis). This is significant because different members of a species complex may have different virulence and spectrum of antifungal resistance.

Suggestions:

Methods: Lines 73-74: the sentence suggests that all the 108 isolates were analysed by phenotypic approach AND molecular methods. However, in the Discussion, the authors say that only 66 of them went trough molecular analysis (line 230). Could the authors explain better if all of the isolates were analysed with molecular methods, or if only a subset of them did? Line 99: follow> follows Line 114: 839 bp should be 850 bp. Line 122: I believe tat the title of this paragraph may be misleading. This method does not allow the identification of the species of the C. haemulonii complex. As a matter of fact, it only allows for the identification of C. auris, as the other members of the complex behave the same way. I think that a title more similar to paragraph 2.5 would be more appropriate. Line 133: I would suggest to change “previous section” to “paragraph 2.2”. Figure 2: the 36 bp band in C. auris profile is missing from the in silico gel. Line 146: The authors should make it clear that they refer to gene QG37_03410. In fact, in the paper that they reference they work with 2 predicted GPI genes. Results: In the entire section the names of the species are not in Italic Figure 3: 188: there is a refuse PC (PC PCR) Line 195: same comments as in line 122 Line 200: have > has Line 205: an specific > a specific; gpi gene: name of the gene

Author Response

Point-by-point response

Reviewer 2

Suggestions:

Methods: Lines 73-74: the sentence suggests that all the 108 isolates were analysed by phenotypic approach AND molecular methods. However, in the Discussion, the authors say that only 66 of them went through molecular analysis (line 230). Could the authors explain better if all of the isolates were analysed with molecular methods, or if only a subset of them did?

Answer: We highly thank the reviewer for noticing this error on our part. The number 66 referred only to the species of the C. albicans complex. The sentence has been changed as follows: “One hundred and eight isolates of the C. albicans, C. glabrata, C. parapsilosis and C. haemulonii complexes were analyzed by molecular methods.”

Line 99: follow> follows

Answer: Corrected. Thank you.

Line 114: 839 bp should be 850 bp.

Answer: Corrected.

Line 122: I believe that the title of this paragraph may be misleading. This method does not allow the identification of the species of the C. haemulonii complex. As a matter of fact, it only allows for the identification of C. auris, as the other members of the complex behave the same way. I think that a title more similar to paragraph 2.5 would be more appropriate.

Answer: We totally agree. The paragraph title has been changed as follows: “2.4. PCR-RFLP for the identification of Candida auris within the C. haemulonii complex”

Line 133: I would suggest to change “previous section” to “paragraph 2.2”.

Answer: Thanks. We have changed this.

Figure 2: the 36 bp band in C. auris profile is missing from the in silico gel.

Answer: The 36 bp band is not missing in the C. auris profile. The reason why it does not appear in the image is that the Geneious software by default does not reflect any band below 100 bp probably because the small bands are also not visible on a conventional agarose gel.

Line 146: The authors should make it clear that they refer to gene QG37_03410. In fact, in the paper that they reference they work with 2 predicted GPI genes.

Answer: The sentence has been modified as follows: “A PCR assay described for the rapid identification of C. auris was also performed on the isolates of the C. haemulonii complex. This technique amplified a unique glycosylphosphatidylinositol (GPI) protein-encoding gene originally described as QG37_03410”

Results: In the entire section the names of the species are not in Italic

Answer: Corrected. Thanks.

Figure 3: 188: there is a refuse PC (PC PCR) Line 195: same comments as in line 122

Answer: Corrected. Thanks.

Line 200: have > has

Answer: Corrected. Thanks.

Line 205: an specific > a specific; gpi gene: name of the gene

Answer: Both were corrected.